# Learning Hierarchical Discrete Linguistic Units from Visually-Grounded Speech

**David Harwath,** * **Wei-Ning Hsu** *, **and James Glass**
Computer Science and Artificial Intelligence Lab
Massachusetts Institute of Technology
Cambridge, MA 02139, USA
`{dharwath,wnhsu,glass}@csail.mit.edu`

## Abstract

In this paper, we present a method for learning discrete linguistic units by incorporating vector quantization layers into neural models of visually grounded speech. We show that our method is capable of capturing both word-level and sub-word units, depending on how it is configured. What differentiates this paper from prior work on speech unit learning is the choice of training objective. Rather than using a reconstruction-based loss, we use a discriminative, multimodal grounding objective which forces the learned units to be useful for semantic image retrieval. We evaluate the sub-word units on the ZeroSpeech 2019 challenge, achieving a 27.3% reduction in ABX error rate over the top-performing submission, while keeping the bitrate approximately the same. We also present experiments demonstrating the noise robustness of these units. Finally, we show that a model with multiple quantizers can simultaneously learn phone-like detectors at a lower layer and word-like detectors at a higher layer. We show that these detectors are highly accurate, discovering 279 words with an F1 score of greater than 0.5.

## 1 Introduction

By 8 months of age, human infants learn to recognize not only the names of their caregivers and common objects, but also the contrast between the different vowels and consonants which comprise these words (Dupoux, 2018). Nearly all toddlers learn to carry a conversation long before they can read and write. Humans learn to model the discrete, hierarchical, and compositional nature of their native language not from written text, but from speech audio - a continuous, time-varying waveform which is the product not only of the underlying words which were spoken, but also the physical properties of the speaker's vocal tract, the speaker's health and emotional state, and the noise and reverberation present in the environment. The question of how such a complex symbolic system is inferred from continuous and noisy sensory input data is of interest not only to the cognitive science community, but also to machine learning researchers who aim to reproduce this ability with computers. A more comprehensive understanding of human language acquisition has practical significance in real-world applications, such as automatic speech recognition (ASR) and natural language understanding (NLU) systems. In the past several decades, enormous progress has been made in speech recognition research, and nowadays ASR systems are able to achieve human-level accuracy in many domains (Chiu et al., 2018). Unfortunately, the techniques that have been developed to achieve these levels of performance are extremely data-hungry, requiring many thousands of hours of speech audio recordings for training. Since supervised machine learning algorithms form the basis of ASR training, the data also needs to be annotated by expert humans. Due to the immense cost of collecting and annotating speech data, ASR technology currently exists for approximately 120 (Google, 2019) out of the nearly 7,000 (Lewis et al., 2016) human languages spoken worldwide. It is highly unlikely that purely supervised machine learning techniques will be able to scale to include all human languages, necessitating the development of alternative methods by researchers which are able to function with far fewer annotations, or even no annotations at all. Because human beings provide an existence proof of language acquisition from speech completely without language supervision, it is plausible that this ability could be replicated by a machine learning algorithm.

---

*Equal contribution

In this paper, we present a method for discovering *discrete* and *hierarchical* representations of speech units both at the sub-word level and the word level. Previously proposed linguistic unit discovery methods have only leveraged the speech audio modality in isolation, relying on objective functions that attempt to capture statistical regularities within the speech signal. The key innovation in our work is that we discover units by training models with explicit discretization layers to associate speech waveforms with visual images using a cross-modal grounding objective. This forces our models to learn representations which capture semantic information at the highest layers of the network. Because semantics are predominantly carried by words, and words are composed of sub-word units (such as phones and syllables), the visual grounding objective indirectly forces the model to learn speaker- and noise-invariant representations of speech units. By incorporating trainable quantization layers into our networks, we are able to capture these units in discrete inventories. Whether these units correspond to word-like or sub-word units depends on where the quantization layers are inserted, and how they are trained.

## 2 RELATED WORK

Prior work on unsupervised modeling of the speech signal has generally focused on learning representations which either disentangle or isolate the latent factors that are of interest for downstream tasks. In most cases the primary latent factor of interest is the phonetic or lexical identity of a given segment of speech, but other factors, such as the identity of the speaker, are sometimes of interest as well. Because the factors of interest are often inherently discrete (e.g. words and phones), many of the proposed approaches attempt to perform segmentation and clustering of the surface features in one way or another. One family of techniques is based upon Segmental Dynamic Time Warping (S-DTW) (Park & Glass, 2005; 2008; Jansen et al., 2010; Jansen & Van Durme, 2011), which uses a self-comparison algorithm to identify relatively long duration (on the order of a second) patterns which frequently reoccur in a speech corpus; these patterns tend to capture words or short phrases. A different line of work employs probabilistic graphical models to jointly segment and cluster the speech signal (Varadarajan et al., 2008; Zhang & Glass, 2009; Gish et al., 2009; Lee & Glass, 2012; Siu et al., 2014; Lee et al., 2015; Ondel et al., 2016; Kamper et al., 2016; 2017a). With an appropriately designed model, it is possible to learn multiple, hierarchical categories of speech units. However, in order to enable efficient inference, the conditional distributions of these models tend to be simple and therefore have limited modeling power.

Deep neural network models have been successfully used to learn powerful speech representations using weakly or unsupervised objectives (Thiolliere et al., 2015; Kamper et al., 2015; Hsu et al., 2017a;b; Hsu & Glass, 2018; Holzenberger et al., 2018; Milde & Biemann, 2018; van den Oord et al., 2018; Chung et al., 2019; Pascual et al., 2019). These representations have predominantly been continuous in nature, as discrete latent variables are not trivially compatible with backpropagation. To obtain discrete representations, a post-hoc clustering step can be applied to the continuous representations (Kamper et al., 2017b; Feng et al., 2019). More recently, several papers have proposed ways of directly incorporating discrete variables into neural network models, including using Gumbel-Softmax (Eloff et al., 2019b) or straight-through estimators (van den Oord et al., 2017; Chorowski et al., 2019; Razavi et al., 2019).

A different method for learning meaningful representations of speech is via a multimodal grounding objective, which encourages the learning of speech representations that are predictive of the contextual information contained in a separate but accompanying modality, such as vision. Visual grounding of speech is a form of self-supervised learning (Virginia de Sa, 1994), which is powerful in part because it offers a way of training models with a discriminative objective that does not depend on traditional transcriptions or annotations. The first work in this direction relied on phone strings to represent the speech (Roy & Pentland, 2002; Roy, 2003), but more recently this learning has been shown to be possible directly on the speech signal (Synnaeve et al., 2014; Harwath & Glass, 2015; Harwath et al., 2016). Subsequent work on visually-grounded models of speech has investigated improvements and alternatives to the modeling or training algorithms (Leidal et al., 2017; Kamper et al., 2017c; Havard et al., 2019a; Merkx et al., 2019; Chrupała et al., 2017; Scharenborg et al., 2018; Kamper et al., 2019b;a; Surís et al., 2019; Ilharco et al., 2019; Eloff et al., 2019a), application to multilingual settings (Harwath et al., 2018a; Kamper & Roth, 2017; Azuh et al., 2019; Havard et al., 2019a), analysis of the linguistic abstractions, such as words and phones, which are learned by the models (Harwath & Glass, 2017; Harwath et al., 2018b; Drexler & Glass, 2017; Alishahi et al.,

2017; Harwath et al., 2019; Harwath & Glass, 2019; Havard et al., 2019b), and the impact of jointly training with textual input (Holzenberger et al., 2019; Chrupała, 2019; Pasad et al., 2019). Representations learned by models of visually grounded speech are also well-suited for transfer learning to supervised tasks, being highly robust to noise and domain shift (Hsu et al., 2019).

# 3    DATA AND MODELS

## 3.1    DATASET

For training our models, we utilize the MIT Places 205 dataset (Zhou et al., 2014) and their accompanying spoken audio captions (Harwath et al., 2016; 2018b). The caption dataset contains approximately 400,000 spoken audio captions, each of which describes a different Places image. These captions are free-form spontaneous speech, collected from over 2,500 different speakers and covering a 40,000 word vocabulary. The average caption duration is approximately 10 seconds, and each caption contains on average 20 words. For vetting our models during training, we use a held-out validation set of 1,000 image-caption pairs.

## 3.2    NEURAL MODELS OF VISUALLY-GROUNDED SPEECH

We base our model upon the Residual Deep Audio-Visual Embedding network (ResDAVEnet) architecture (Harwath et al., 2019), which contains two branches of fully convolutional networks, one for images and the other for audio. Each branch encodes samples of the corresponding modality into a $d$-dimensional space, regardless of the original dimensionality of the samples. This is achieved by applying global spatial mean pooling and global temporal mean pooling to the image branch output and the audio branch output, respectively. The image branch is adapted from ResNet50 (He et al., 2016), where the final softmax layer and the preceding fully-connected layers are removed, replaced with a 1x1 linear convolutional layer in order to project the feature map to the desired dimension. To model the audio inputs, a 17-layer fully convolutional network with residual connections is used. The input is a log Mel-frequency spectrogram with 40 frequency bins and 25 ms-wide, Hamming-windowed frames with a shift of 10 ms. The first layer of this network is a 1-D convolution that spans the entire frequency axis of the spectrogram, while the remaining 16 convolutional layers are 1-D across the time axis. These 16 layers are divided into four residual blocks of 4 layers each, and downsampling between these blocks is accomplished by applying the first convolution of each block with a stride of 2. For full details of the model, refer to Harwath et al. (2019).

## 3.3    LEARNING HIERARCHICAL DISCRETE UNITS WITH VECTOR QUANTIZING LAYERS

Previous analyses reveal that ResDAVEnet-like models learn linguistic abstractions at different levels, including words (Harwath & Glass, 2017) and robust phonetic features (Harwath & Glass, 2019; Hsu et al., 2019). To explicitly learn hierarchical discrete linguistic units within this framework, we propose to incorporate multiple vector quantization (VQ) layers (van den Oord et al., 2017) into the ResDAVEnet audio branch; we refer to this new architecture as ResDAVEnet-VQ.

VQ layers can be understood as a type of bottleneck, which constrain the amount of information that can flow through. While these layers have been used to learn discrete sub-word units (van den Oord et al., 2017; Chorowski et al., 2019; Razavi et al., 2019), previous work injects VQ layers into autoencoders that are trained with a reconstruction loss. As a result, the embedding dimension of each code and the number of codes need to be carefully tuned (Liu et al., 2019). When the embedding dimension is too low or the codebook size too small, the model does not have enough expressive power to capture linguistic variability. When it is too large, the model starts to encode non-linguistic information in order to improve reconstruction. In contrast, the learning signal of ResDAVEnet-VQ is provided by the visual-semantic grounding objective. Rather than encoding as much information about input as possible, the learned codes in ResDAVEnet-VQ only need to capture semantic information. Since semantics in speech are predominantly transmitted by words, and words are composed of sub-word units like phones, the grounding objective places pressure on the model to robustly infer both from speech. Since words and phones are inherently discrete symbols, representing them with learned discrete units may not even hurt the grounding performance.

Figure 1 illustrates the proposed ResDAVEnet-VQ model. We add a quantization layer after each of the first two residual blocks of the ResDAVEnet-VQ model, denoted as VQ2 and VQ3, respectively, with the intention that they should capture discrete sub-word-like and word-like units. A VQ layer is defined as $\boldsymbol{E} \in \mathbb{R}^{K \times D}$, where $K$ represents the codebook size, and $D$ represents the output dimensionality of the input features to the codebook. Denoting the $t^{th}$ temporal frame of the input to the quantization layer as $\boldsymbol{x}_t$, quantization is performed according to $\boldsymbol{q}_t = \boldsymbol{E}_{k,:}$, where $k = \arg\min_j ||\boldsymbol{x}_t - \boldsymbol{E}_{j,:}||_2$ The quantized output is then fed as input to the subsequent residual block. As in van den Oord et al. (2017), we use the straight-through estimator (Bengio et al., 2013) to compute the gradient passed from $\boldsymbol{q}_t$ to $\boldsymbol{x}_t$. We use the exponential moving average (EMA) codebook updates proposed by van den Oord et al. (2017).

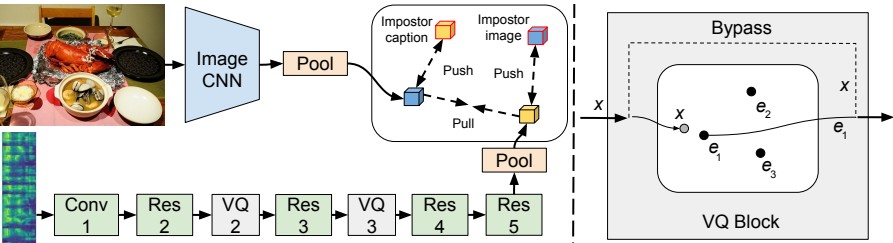

Figure 1: Diagram of the ResDAVEnet-VQ model. On the left, we show the placement of the vector quantization blocks in the audio branch. Note that each "Res" block is comprised of a stack of multiple sub-layers (see Harwath et al. (2019) for details). The right half of the figure depicts the quantization mechanism of each VQ block, as well as the bypass path when the block is disabled.

## 3.4 Codebook Learning Schedules

We include multiple VQ layers in the ResDAVEnet-VQ model, each of which can be independently enabled or bypassed without changing the rest of the architecture configuration. When all model weights, including the VQ codebooks, are trained jointly in a single training run we call this a "cold-start" model. Alternatively, a model can be "warm-started" by copying the weights from another trained model that has fewer (or no) VQ layers enabled, and randomly initializing the codebook of the newly activated VQ layer(s). This gives rise to the questions of how many quantizers should be used and in what order they should be enabled. It is unclear whether models with the same VQ layers activated would learn the same representation at each layer regardless of the training curriculum. Let $A_m$ denote a subset of all VQ layers, and $A_{m-1} \subset A_m$. We use "$A_1 \to ... \to A_M$" to denote a model that is obtained by sequentially training models "$A_1 \to ... \to A_m$" initialized from "$A_1 \to ... \to A_{m-1}$", where the model $A_1$ is initialized from scratch, and the final model would have VQ layers in $A_M$ activated. For instance, a model initialized from scratch with no VQ layers enabled is denoted as "$\varnothing$", and a model initialized with that and with both layers enabled is denoted as "$\varnothing \to \{2, 3\}$".

## 3.5 Training with the Triplet Loss

We train our models using the same loss function as Harwath et al. (2019). This loss function blends two triplet loss terms (Weinberger & Saul, 2009), one based on random sampling of negative examples, and the other based on semi-hard negative mining (Jansen et al., 2018), in order to find more challenging negative samples. Specifically, let the sets of output embedding vectors for a minibatch of $B$ audio/image training pairs respectively be $\mathbb{A} = \{\boldsymbol{a}_1, \ldots, \boldsymbol{a}_B\}$ and $\mathbb{I} = \{\boldsymbol{i}_1, \ldots, \boldsymbol{i}_B\}$. To compute the randomly-sampled triplet loss term, we select impostor examples for the $j^{th}$ input according to $\bar{\boldsymbol{a}}_j \sim \text{UniformCategorical}(\{\boldsymbol{a}_1, \ldots, \boldsymbol{a}_B\} \backslash \boldsymbol{a}_j)$ and $\bar{\boldsymbol{i}}_j \sim \text{UniformCategorical}(\{\boldsymbol{i}_1, \ldots, \boldsymbol{i}_B\} \backslash \boldsymbol{i}_j)$. The randomly-sampled triplet loss is then computed as:

$$\mathcal{L}_s = \sum_{j=1}^{B} \Big( \max(0, \boldsymbol{i}_j^T \bar{\boldsymbol{a}}_j - \boldsymbol{i}_j^T \boldsymbol{a}_j + 1) + \max(0, \bar{\boldsymbol{i}}_j^T \boldsymbol{a}_j - \boldsymbol{i}_j^T \boldsymbol{a}_j + 1) \Big) \tag{1}$$

For the semi-hard negative triplet loss, we first define the sets of impostor candidates for the $j^{th}$ example as $\hat{\mathbb{A}}_j = \{\boldsymbol{a} \in \mathbb{A} | \boldsymbol{i}_j^T \boldsymbol{a} < \boldsymbol{i}_j^T \boldsymbol{a}_j\}$ and $\hat{\mathbb{I}}_j = \{\boldsymbol{i} \in \mathbb{I} | \boldsymbol{i}^T \boldsymbol{a}_j < \boldsymbol{i}_j^T \boldsymbol{a}_j\}$. The semi-hard negative

loss is then computed as:

$$\mathcal{L}_h = \sum_{j=1}^{B} \Big( \max(0, \max_{\hat{\boldsymbol{a}} \in \hat{\mathbb{A}}_j}(\boldsymbol{i}_j^T \hat{\boldsymbol{a}}) - \boldsymbol{i}_j^T \boldsymbol{a}_j + 1) + \max(0, \max_{\hat{\boldsymbol{i}} \in \hat{\mathbb{I}}_j}(\hat{\boldsymbol{i}}^T \boldsymbol{a}_j) - \boldsymbol{i}_j^T \boldsymbol{a}_j + 1) \Big) \tag{2}$$

Finally, the overall loss function is computed by combining the two above losses, $\mathcal{L} = \mathcal{L}_s + \mathcal{L}_h$, which was found by (Harwath et al., 2019) to outperform either loss on its own.

### 3.6 IMPLEMENTATION DETAILS

All of our models were trained for 180 epochs using the Adam optimizer (Kingma & Ba, 2014) with a batch size of 80. We used an exponentially decaying learning rate schedule, with an initial value of 2e-4 that decayed by a factor of 0.95 every 3 epochs. Following van den Oord et al. (2017), we use an EMA decay factor of $\gamma = .99$ for training each VQ codebook. Our core experimental results all use a codebook size of 1024 vectors for all quantizers, but in the supplementary material we include experiments with smaller and larger codebooks. Following Chorowski et al. (2019), the jitter probability hyperparameter for each quantization layer was fixed at 0.12. While we do not apply data augmentation to the input spectrograms, during training we perform standard data augmentation techniques to the images. We resize each raw image so that its smallest dimension is 256 pixels, and then we apply an Inception-style random crop which is resized to 224 pixels square. During training, we also flip each image horizontally with a probability of 0.5. During evaluation, the center 224 pixel square crop is always taken from the image. Finally, the RGB pixel values are mean and variance normalized. We trained each model on the Places audio caption train split, and computed the image and caption recall at 10 (R@10) scores on the validation split of the Places audio captions after each training epoch. The model snapshot that achieved the highest average R@10 score on the validation set from each training is used for all evaluation. To extract embeddings and units from our models, we simply perform a forward pass through the speech branch of the ResDAVEnet-VQ network and retain the outputs from the target layer at a uniform frame-rate. The frame-rate is determined by the downsampling factor at the target layer relative to the input. For non-quantized layers, these outputs will be continuous embeddings. For quantized layers, these will be quantized embedding retrieved from the assigned entry in the codebook.

## 4 EXPERIMENTS

### 4.1 SUB-WORD UNIT LEARNING ON THE ZEROSPEECH 2019 ABX TASK

**Evaluation metrics** Learning unsupervised speech representations that are indicative of phonetic content is of high interest to the speech community, and recently has been the focus of the ZeroSpeech Challenge (Versteegh et al., 2015; Dunbar et al., 2017; 2019). One of the core evaluations is the minimal-pair ABX task (Schatz et al., 2013), which aims to benchmark representations in terms of their discriminability between different sub-word speech units. In this task, a model is tasked with extracting representations for a triplet of speech waveform segments denoted by $A$, $B$, and $X$. $A$ and $B$ are constrained to be a triphone minimal pair; that is, both segments capture three phones, but differ only in the identity of their center phone. The third segment, $X$ is chosen to contain the same underlying triphone sequence as $A$. Supposing $f(\cdot)$ denotes the model's mapping function from a waveform segment to a sequence of embedding vectors, the ABX error rate under a given similarity metric $S(\cdot, \cdot)$ is defined as the fraction of ABX triples in which $S(f(A), f(X)) > S(f(B), f(X))$. An ABX error rate of 50% indicates random assignment, while an ABX of 0% reflects perfect phone discriminability. In the ZeroSpeech challenge, $S(\cdot, \cdot)$ is implemented using Dynamic Time Warping (DTW) with various distance measures (cosine, KL, etc.). In our evaluation, we use the cosine distance.

The ZeroSpeech 2019 challenge in particular emphasizes on discovering an inventory of discrete sub-word units, rather than continuous representations. Therefore, in addition to an ABX error rate, a bitrate is also computed for each model which reflects the amount of information carried by the learned units. A lower bitrate can be achieved by having a more compact inventory of learned units or having a smaller number of codes per second. The full details of the evaluation can be found in Dunbar et al. (2019). To be clear, all of our ResDAVEnet-VQ models were not trained on the

Table 1: Comparison of R@10, ABX scores, and bit-rates between different configurations and baseline models trained on ZeroSpeech 2019 data or Places Audio Caption. All quantizers reflected in this table used a codebook size of 1,024 vectors. We do not compute RLE or segment scores for the FHVAE-DPGMM model, since we did not re-implement that model.

| Model ID | Layer | R@10 | Frame-Based | | | Segment-Based | |
| --- | --- | --- | --- | --- | --- | --- | --- |
| | | | ABX | Bitrate | RLE Bitrate | ABX | Bitrate |
| FHVAE-DPGMM (ZS) | N/A | N/A | 21.67 | 413.23 | - | - | - |
| WaveNet-VQ (ZS) | N/A | N/A | 19.98 | 151.55 | 136.74 | 20.48 | 126.17 |
| WaveNet-VQ (PA) | N/A | N/A | 24.87 | 149.00 | 136.27 | 25.23 | 126.22 |
| "∅" | Res2 | .735 | 11.35 | N/A | N/A | N/A | N/A |
| | Res3 | | 10.86 | N/A | N/A | N/A | N/A |
| "{2}" | VQ2 | .753 | 12.33 | 433.30 | 361.09 | 12.78 | 332.86 |
| "∅ → {2}" | VQ2 | .760 | **11.79** | 390.61 | 317.66 | 12.66 | 289.11 |
| "{3}" | VQ3 | .734 | 38.21 | 213.92 | 129.65 | 38.68 | 108.84 |
| "∅ → {3}" | VQ3 | .794 | **15.04** | 182.93 | 140.04 | 16.53 | 121.26 |
| "{2, 3}" | VQ2 | .667 | 25.62 | 408.75 | 258.37 | 26.32 | 217.58 |
| | VQ3 | | 32.23 | 218.76 | 156.69 | 32.49 | 136.90 |
| "∅ → {2, 3}" | VQ2 | .787 | 13.15 | 405.43 | 334.39 | 13.30 | 303.03 |
| | VQ3 | | 14.95 | 199.91 | 172.05 | 15.60 | 159.07 |
| "{2} → {2, 3}" | VQ2 | .764 | **12.51** | 415.13 | 341.85 | 13.06 | 311.82 |
| | VQ3 | | **14.52** | 167.84 | 136.11 | 15.68 | 121.17 |
| "{3} → {2, 3}" | VQ2 | .760 | 13.55 | 421.23 | 271.91 | 14.38 | 232.87 |
| | VQ3 | | 33.70 | 208.63 | 117.37 | 33.58 | 98.29 |

ZeroSpeech training data, but instead on the Places audio captions, thus there is a domain mismatch between training and testing these models.

In addition to the frame-based bitrate and ABX scores computed by the ZeroSpeech 2019 evaluation toolkit, we implement our own extensions to these metrics. Because it is common for successive frames to be assigned to the same codebook entry and phonetic information is not encoded at a fixed frame rate, lossless run length encoding (RLE) can be a more reasonable measure of the bitrate of a frame-based model. RLE does not change the ABX score since it can be trivially inverted, but it does change the bitrate. For computing the RLE bitrate, we modify the bitrate calculation specified in Dunbar et al. (2019) so that a unique symbol is defined as the tuple (unit, length) where length is the number of frames assigned to a given unit with in a segment. We also consider segment-based ABX and bitrate, which is similar to the RLE metrics except in this case we outright discard the frame length information. This typically results in an even greater reduction in bitrate, but also an accompanying deterioration in ABX score.

**Baseline models** In Table 1, we compare our results to those derived from two of the top-performing submissions to the ZeroSpeech 2019 challenge: a re-implementation of WaveNet-VQ (Chorowski et al., 2019) provided by Cho et al. (2019) and FHVAE-DPGMM (Feng et al., 2019). Using the code accompanied with the WaveNet-VQ submission, we were able to train their model on the set of 400,000 Places audio captions to make a fairer comparison with our ResDAVEnet-VQ models in terms of the amount of speech data used. In addition, when trying to reproduce the reported WaveNet-VQ results, we obtain better performance than previously reported by training for more steps. Table 1 shows that WaveNet-VQ achieves similar bitrates regardless of the training data. However, ABX deteriorates from 19.98 to 24.87, implying the model cannot utilize data of a larger scale but out-of-domain relative to the test set. A similar degradation when testing on out-of-domain data with FHVAE models was observed in Hsu et al. (2019). We did not re-train the model submitted by Feng et al. (2019), and instead compare against the scores reported in Dunbar et al. (2019).

**ABX discrimination without using quantization** Our first experiment investigates exactly which layer in the ResDAVEnet-VQ model is most suited for ABX phone discrimination, and would thus make a good candidate for learning of quantized sub-word units. The leftmost plot in Figure 2 shows that layers 2 and 3 of a ResDAVEnet-VQ model without any quantization enabled perform

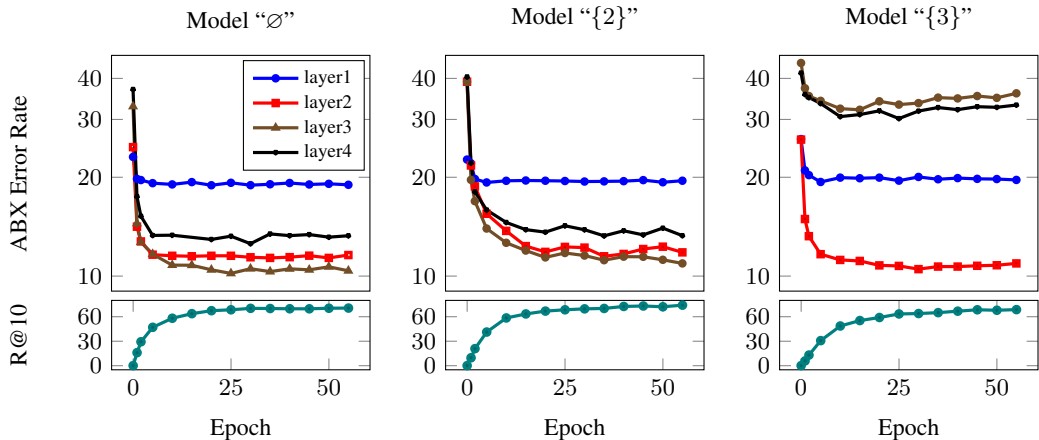

Figure 2: R@10 and ABX tracked at various training epochs. The "∅" model achieves a final R@10 of .735, with ABX scores of 19.77, 11.35, 10.86, and 14.05 for the conv1, res2, res3, and res4 layers.

the best in terms of ABX error rate on the ZeroSpeech 2019 English test set; the exact numbers for this model are displayed in the caption of Figure 2. Because layers 2 and 3 achieve the lowest ABX error rates without quantization, we focus our attention on the impact of quantization there.

**Quantizing one layer** When quantizing only one layer, we examine quantization of layer 2 vs. layer 3, and using cold-start training vs. warm-start initialization from model "∅". The ABX and bitrate results for these models, as well as the R@10 scores on the Places validation set, are shown in Table 1. In all cases, quantization applied at the output of layer 2 achieves a better ABX score than quantization at layer 3, but VQ3 achieves a better bitrate. Quantization barely impacts the performance of layer 2, whose ABX score very slightly rises from 11.35 to 11.79. Warm-start initialization is beneficial to R@10 and ABX score in both cases, but we notice an intriguing anomaly when applying cold-start quantization to layer 3: the ABX score deteriorates significantly, rising from 10.86 in the case of the non-quantized model to 38.21. This indicates that while VQ2 is capable of learning a finite inventory of units that are highly predictive of phonetic identity from either a warm-start or cold-start initialization, cold-start training of VQ3 results in very little phonetic information captured by the quantizer. Interestingly, this model is still learning to infer visual semantics from the speech signal, as evidenced by a high R@10 score; we later show in Section 4.2 that the reason for this anomaly is because cold-start training of VQ3 results in the learning of word detectors. In all cases except for model "{3}", we note that the ABX scores achieved by our models are significantly better than the baselines. Our best model in terms of ABX ("∅ → {2}") achieves a 41.0% reduction in ABX over the WaveNet-VQ baseline, at a cost of a 132.3% increase in RLE bitrate; however, model "∅ → {3}" achieves a 24.7% reduction in ABX error rate with only a 2.4% increase in RLE bitrate. These results do not constitute a fair comparison, however, because the WaveNet-VQ and ResDAVEnet-VQ models were trained on different datasets; when training the WaveNet-VQ model on the same set of audio captions used to train ResDAVEnet-VQ (but without the accompanying images, since WaveNet-VQ is not a multimodal model), the ABX error rate increases to 24.87%, tipping the results even more in favor of the ResDAVEnet-VQ models.

**Quantizing two layers** Quantizing multiple layers at once offers the possibility of learning a hierarchy of units. Thus, we aim to capture phonetic information in a lower layer quantizer and word-level information at a higher layer quantizer. Cold-start training of two quantizers ("{2, 3}") results in a significant drop in ABX performance for both VQ2 and VQ3, but also a drop in R@10 on the Places validation set. We see much better results in terms of R@10 and ABX for the remaining 3 models which were initialized from the "∅" model or a model with only one quantizer enabled; for example, model "{2} → {2, 3}" achieves an ABX of 14.52 with an RLE bitrate of 136.11, representing a 27.3% ABX improvement over the best baseline while keeping the bitrate approximately the same. We see in model "{3} → {2, 3}" that the same phenomenon observed with model "{3}" persists: VQ3 achieves relatively poor ABX, despite a high overall R@10 and strong ABX with VQ2 at 13.55%. We confirm in Section 4.2 that the VQ3 layer of model "{3} → {2, 3}" does

indeed capture word-level information, indicating that this model has successfully localized phonetic unit identity in the second layer and lexical unit identity in the third layer. Overall, our results suggest that when learning hierarchical quantized representations with a ResDAVEnet-VQ model, the nature of the representations learned is highly dependent on the training curriculum.

Table 2: ABX scores and RLE bitrates for various SNRs on the noisy ZeroSpeech19 English test set. "R-B" stands for "RLE-Bitrate," and (n) denotes a model trained on the noisy Places Audio dataset. For the WaveNet-VQ models, (ZS) and (PA) respectively denote training on the ZeroSpeech 19 English training set, and the clean Places Audio dataset.

| Model | Layer | Clean | | 20-30 dB | | 10-20 dB | | 0-10 dB | |
|---|---|---|---|---|---|---|---|---|---|
| | | ABX | R-B | ABX | R-B | ABX | R-B | ABX | R-B |
| WaveNet-VQ (ZS) | N/A | 19.98 | 136.74 | 21.22 | 141.07 | 27.51 | 144.28 | 42.55 | 126.96 |
| WaveNet-VQ (PA) | N/A | 24.87 | 136.27 | 27.18 | 137.70 | 33.29 | 132.34 | 42.67 | 110.50 |
| "$\varnothing$" | Res2 | 11.35 | N/A | 11.63 | N/A | 13.17 | N/A | 19.44 | N/A |
| "$\varnothing$" | Res3 | 10.86 | N/A | 11.16 | N/A | 12.96 | N/A | 19.43 | N/A |
| "$\varnothing \to \{2\}$" | VQ2 | 11.79 | 317.66 | 12.15 | 325.40 | 14.62 | 332.21 | 23.96 | 327.15 |
| "$\{2\} \to \{2,3\}$" | VQ2 | 12.51 | 341.85 | 12.56 | 350.28 | 14.82 | 362.73 | 25.02 | 330.54 |
| "$\{2\} \to \{2,3\}$" | VQ3 | 14.52 | 136.11 | 14.73 | 137.68 | 17.44 | 143.14 | 27.68 | 133.13 |
| "$\{3\} \to \{2,3\}$" | VQ2 | 13.55 | 271.91 | 13.65 | 272.46 | 15.69 | 267.70 | 24.06 | 244.52 |
| "$\{3\} \to \{2,3\}$" | VQ3 | 33.70 | 117.37 | 32.56 | 118.22 | 34.65 | 115.40 | 39.82 | 102.48 |
| "$\varnothing$" (n) | Res2 | 13.32 | N/A | 12.30 | N/A | 12.97 | N/A | 16.91 | N/A |
| "$\varnothing$" (n) | Res3 | 11.85 | N/A | 11.90 | N/A | 12.44 | N/A | 16.09 | N/A |
| "$\varnothing \to \{2\}$" (n) | VQ2 | 12.64 | 342.53 | 12.20 | 348.57 | 13.34 | 359.43 | 18.82 | 373.60 |
| "$\{2\} \to \{2,3\}$" (n) | VQ2 | 13.42 | 365.89 | 13.71 | 359.14 | 14.57 | 370.67 | 18.78 | 392.10 |
| "$\{2\} \to \{2,3\}$" (n) | VQ3 | 14.39 | 179.19 | 14.92 | 180.36 | 15.38 | 182.27 | 19.58 | 188.32 |
| "$\{3\} \to \{2,3\}$" (n) | VQ2 | 16.52 | 223.28 | 16.47 | 223.61 | 17.75 | 225.72 | 22.68 | 230.01 |
| "$\{3\} \to \{2,3\}$" (n) | VQ3 | 26.21 | 187.31 | 25.88 | 187.92 | 26.34 | 188.49 | 31.26 | 191.28 |

**Training and testing on noisy data**  In Hsu et al. (2019), it was shown that representations learned by a ResDAVEnet model were far more robust to train/test domain mismatch in terms of background noise, channel characteristics, and speaker identity than standard spectral features when training a supervised speech recognizer. Here, we examine whether this robustness is also exemplified by the quantized versions of this model. We construct three additional test sets using the ZeroSpeech 2019 English testing data by adding noise sampled from the AudioSet (Jansen et al., 2018) dataset. For each ZeroSpeech testing waveform, we randomly sampled an AudioSet waveform of the same duration and performed linear mixing with a signal-to-noise ratio (SNR) selected randomly within a specified range. We construct low, medium, and high noise testing sets, corresponding to SNRs of 20-30 dB, 10-20 dB, and 0-10 dB. We then perform the ABX discrimination task on these noisy waveforms, displaying the results in Table 2. We find that for all models, a worsening SNR results in a deterioration in ABX performance. However, the ResDAVEnet-VQ models prove to be far more noise robust than the Wavenet-VQ model; even in the high noise testing set, the best ResDAVEnet-VQ model achieves an ABX of 23.96%, while the WaveNet-VQ models degrade to nearly-random ABX scores of 42.55% and 42.67%.

Given that a ResDAVEnet-VQ model trained on the "clean" Places Audio captions is highly robust to additive noise on the ABX discrimination task, we investigated whether adding noise to the Places Audio captions themselves would result in an even higher degree of noise robustness. To that end, we followed a similar data augmentation approach to create a noisy version of the Places Audio captions, where the SNR of each caption was randomly chosen to sit within the range of 0-30 dB. The bottom half of Table 2 shows the results of training several ResDAVEnet-VQ models on the noisy Places Audio captions and testing on the clean and noisy ZeroSpeech ABX tasks. In general, we observe a degradation ABX score in the clean conditions, but with a significantly higher degree of noise robustness in the noisier conditions.

**Visualization of learned units**  To better measure the correspondence between the VQ units and English phones, we compute corpus-level co-occurrence statistics (at the frame-level) across the TIMIT training set, excluding the `sa` dialect sentences. To facilitate visualization, we use the "$\varnothing \to \{2\}$" model with a codebook size of 128. We display the conditional probability matrix $P(phone|unit)$ in Figure 3, with the rows and columns ordered via spectral co-clustering with 10

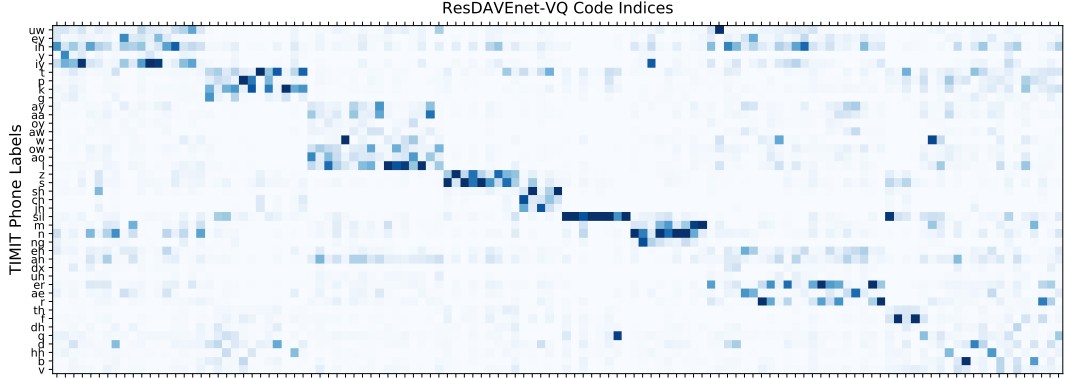

Figure 3: Conditional probability matrix displaying $P(phone|unit)$ using the "$\varnothing \to \{2\}$" model with a VQ2 codebook size of 128. For visualization, we saturate the color scaling at probability 0.5.

clusters in order to group together phones that share similar sets of VQ codes. Visually, there is a strong mapping between TIMIT phone labels and ResDAVEnet-VQ codes. In some cases, redundant codes are used for the same phone label (this is especially the case for the silence label), and in other cases we see that phones belonging to the same manner class often tend to share codebook units. We can numerically quantify the mapping between the phone and unit labels with the normalized mutual information measure (NMI), which we found to be .378 in this case. We also include several caption spectrograms with their time-aligned unit sequences in Figures 5, 6, and 7 in the supplementary material.

Table 3: Performance of the VQ3 layer from the "$\{3\} \to \{2, 3\}$" model when codes are treated as word detectors. Codes are ranked by the highest F1 score among the retrieved words for a given code. Word hypotheses for a given code are ranked by the F1 score. P denotes precision, R recall, and occ the number of co-occurrences of the code and word in the data.

| rank | code | Top Hypotheses | | | | | Second Hypotheses | | | | |
|---|---|---|---|---|---|---|---|---|---|---|---|
| | | word | F1 | P | R | occ | word | F1 | P | R | occ |
| 1 | 918 | pantry | 90.67 | 88.29 | 93.18 | 41 | spice | 3.96 | 2.20 | 20.00 | 1 |
| 2 | 596 | kitchen | 90.08 | 91.59 | 88.63 | 304 | countertop | 1.64 | 0.84 | 29.63 | 8 |
| 3 | 88 | classroom | 88.97 | 89.05 | 88.89 | 72 | classrooms | 5.01 | 2.57 | 100.00 | 2 |
| 4 | 58 | baseball | 88.71 | 88.63 | 88.78 | 182 | player | 3.01 | 1.65 | 17.11 | 13 |
| 5 | 706 | background | 87.86 | 91.93 | 84.14 | 838 | ground | 0.58 | 0.39 | 1.18 | 4 |
| | | | | | $\cdots$ | | | | | | |
| 198 | 237 | lobby | 68.43 | 56.77 | 86.11 | 31 | waiting | 9.93 | 7.86 | 13.46 | 14 |
| 199 | 829 | shirt | 68.41 | 71.49 | 65.58 | 322 | shirts | 18.28 | 10.37 | 76.79 | 43 |
| 200 | 59 | grass | 68.31 | 56.53 | 86.28 | 503 | grassy | 15.30 | 8.67 | 65.35 | 83 |

## 4.2 FROM PHONES TO WORDS: LEARNING A HIERARCHY OF UNITS

As shown in Table 1, all of the ResDAVEnet-VQ models which underwent cold-start training of VQ3 exhibited a similar phenomenon in which the ABX error rate of that layer was particularly high, despite the model performing well at the image-caption retrieval task. We hypothesized that this could be due to VQ3 learning to recognize higher level linguistic units, such as words. To examine this empirically, we inferred the VQ3 unit sequence for every audio caption in the Places Audio training set according to several different models. Using the estimated word-level transcriptions of the utterances (provided by the Google SpeechRecognition API), we computed precision, recall, and F1 scores for every unique (word, VQ3 code) pair for a given model and quantization layer. We then ranked the VQ codes in descending order according to their maximum F1 score for any word in the vocabulary. Table 3 shows a sampling of these statistics for model "$\{3\} \to \{2, 3\}$". In the supplementary material, we include many more examples for this model in Table 7, as well as

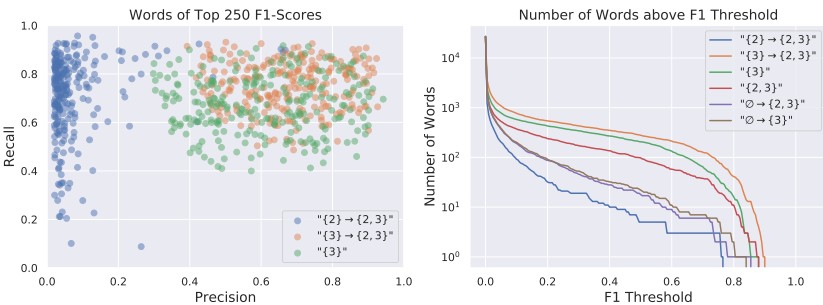

Figure 4: Visualization of the precision, recall, and F1 scores of individual VQ3 codes when treated as word detectors on the Places Audio captions.

examples for the "$\{2\} \to \{2, 3\}$" model (which did *not* learn VQ3 word detectors) in Table 8. It should be emphasized that these models are exactly the same in all respects, except for the order in which their quantizers were trained.

We examine the overall performance of VQ3 as a word detector for these models in Figure 4. The right hand side of Figure 4 displays the number of VQ3 codes whose maximum F1 score is above a given threshold, while the left hand side shows the distribution of precision and recall scores for the top 250 words ranked by F1. This gives an approximate indication of how many VQ3 codes have learned to specialize as detectors for a specific word. We see that the VQ3 layer of model "$\{3\} \to \{2, 3\}$" learns 279 codebook entries with an F1 score above 0.5. In contrast, the VQ3 layer of model "$\{2\} \to \{2, 3\}$" learns only a handful of word-detecting codebook entries with an F1 of greater than 0.5. This experiment supports the notion that the reason for the poor ABX performances of the VQ3 layer in models "$\{3\}$" and "$\{3\} \to \{2, 3\}$" is in fact due to its specialization for detecting specific words, and that this specialization only emerges when the VQ3 layer is learned before the VQ2 layer. Section A.2 in the supplementary material examines this phenomenon in greater experimental detail.

## 5    CONCLUSIONS

In this paper, we demonstrated that the neural vector quantization layers proposed by van den Oord et al. (2017) can be integrated into the visually-grounded speech models proposed by Harwath et al. (2019). This resulted in the ability of the speech model to directly represent speech units, such as phones and words, as discrete latent variables. We presented extensive experiments and analysis of these learned representations, demonstrating significant improvements in phone discrimination ability over the current state-of-the-art models for sub-word speech unit discovery. We demonstrated that these units are also far more robust to noise and domain shift than units derived from previously proposed models. These results supported the notion that semantic supervision via a discriminative, multimodal grounding objective has the potential to be more powerful than reconstruction-based objectives typically used in unsupervised speech models.

We also showed how multiple vector quantizers could be employed simultaneously within a single ResDAVEnet-VQ model, and that these quantizers could be made to specialize in learning a hierarchy of speech units: specifically, phones in the lower quantizer and words in the upper quantizer. Our analysis showed that hundreds of codebooks in the upper quantizer learned to perform as word detectors, and that these detectors were highly accurate. Our experiments also revealed that this behavior only emerged when VQ3 was trained before VQ2. These results suggest the importance of the learning curriculum, which should be more deeply investigated in future work. Future work should attempt to make explicit what kind of compositional rules are implicitly encoded by these models when mapping sequences of codes from the lower quantizer to word-level units in the upper quantizer; the automatic derivation of a sub-word unit inventory, vocabulary, and pronunciation lexicon could serve as the starting point for a fully unsupervised speech recognition system. Future work should also investigate whether layers above VQ3 could be made to learn even higher-level linguistic abstractions, such as grammar, syntax, and compositional reasoning.

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

# A APPENDIX

## A.1 VARYING THE CODEBOOK SIZE.

In Table 4, we examine the impact of varying the codebook size of model "$\varnothing \to \{2\}$" from 128 through 2048. We find that the ABX score is best for 1024 codebook vectors, although the performance is quite good for all models. Unsurprisingly, models with smaller codebooks also achieve lower bitrates.

Table 4: ABX scores and bitrates for various codebook sizes on the clean ZeroSpeech19 English test set, using the "$\varnothing \to \{2\}$" model.

| Codebook size | R@10 | ABX | Bitrate | RLE-Bitrate | Segment-ABX | Segment-Bitrate |
|---|---|---|---|---|---|---|
| 128 | .772 | 14.25 | 295.65 | 212.27 | 15.42 | 179.38 |
| 256 | .756 | 12.95 | 341.18 | 260.10 | 14.21 | 228.07 |
| 512 | .761 | 12.59 | 363.95 | 288.64 | 13.10 | 259.94 |
| 1024 | .760 | 11.79 | 390.61 | 317.66 | 12.66 | 289.11 |
| 2048 | .768 | 12.41 | 360.04 | 283.68 | 13.15 | 254.23 |

## A.2 THE IMPACT OF THE VQ TRAINING CURRICULUM ON THE LOCALIZATION OF WORD DETECTORS

In Section 4.2, we showed that cold-start training of the VQ3 layer caused its codebook vectors to specialize as word detectors, whereas warm-start training did not. Our subsequent experiments (Table 6) revealed that when adding a third quantization layer at the Res4 position to a model that did not learn word detectors at VQ3, the VQ4 layer did in fact learn many word detectors. This suggests implicit word recognition ability can be localized at different layers in the ResDAVEnet audio model, and exactly where it emerges depends upon the VQ training curriculum. We hypothesize that this is due in part to two factors:

Table 5: ABX scores on the ZeroSpeech 19 English test set using features derived from the output of the Res3 block of the ResDAVEnet audio branch (pre-quantization).

| Model ID | Res3 ABX |
|---|---|
| "$\varnothing$" | 10.86 |
| "$\varnothing \to \{2\}$" | 11.61 |
| "$\varnothing \to \{3\}$" | 10.91 |
| "$\varnothing \to \{2, 3\}$" | 12.68 |
| "$\{2\}$" | 11.45 |
| "$\{2\} \to \{2, 3\}$" | 11.37 |
| "$\{3\}$" | 32.24 |
| "$\{3\} \to \{2, 3\}$" | 28.33 |

1. In a warm-start model, whatever type of information (subword-like, word-like) the continuous model learned to encode at a particular convolutional layer (or residual block) does not change after a quantizer is appended to that layer.

2. In a cold-start model, each active quantization layer forms a potential bottleneck, restricting the amount of information that is able to pass through to subsequent layers.

According to this hypothesis, if word-level recognition tends to emerge at a particular layer in an unconstrained network with no quantization bottleneck, it will stay there when quantization is introduced for fine-tuning. However, when a quantization bottleneck is introduced from the very beginning of training, the gradient flowing down into the lower network layers is more constrained during the initial training epochs (when the gradient tends to be the largest). This may have the effect of steering the optimizer into a different part of the parameter space, in which word recognition occurs at a different layer than it otherwise would.

We present results from two experiments that support this view. In Table 5, we show the ABX scores of the Res3 layer prior to quantization (if present) during the course of three different training curricula resulting in a "$\{2, 3\}$" final model. We observe that the ABX error rate changes very little within each individual curriculum. This indicates that the initial model sets the stage for which layer learns to capture phonetic information with the highest salience, and that subsequent training steps do not tend to move this information elsewhere.

Table 6: The number of codebook vectors at a particular VQ layer that learned to be a detector for any word with an F1 score greater than 0.5.

| # Quantized Layers | Model ID | VQ Layer | # Word Detectors (F1 > 0.5) |
|---|---|---|---|
| 1 | "{3}" | 3 | 210 |
| | "$\varnothing \to \{3\}$" | 3 | 20 |
| 2 | "{2, 3}" | 2 | 93 |
| | | 3 | 100 |
| | "$\varnothing \to \{2, 3\}$" | 2 | 1 |
| | | 3 | 18 |
| | "$\{2\} \to \{2, 3\}$" | 2 | 0 |
| | | 3 | 5 |
| | "$\{3\} \to \{2, 3\}$" | 2 | 32 |
| | | 3 | 279 |
| 3 | "$\{2\} \to \{2, 3\} \to \{2, 3, 4\}$" | 2 | 0 |
| | | 3 | 8 |
| | | 4 | 253 |

Table 6 displays the number of word detectors learned by various VQ layers across different training curricula. Here, we claim that a codebook vector belonging to a particular VQ layer has learned to be a word detector if its F1 score for any word appearing in the test set exceeds 0.5 (as measured on the test set). There are several interesting things to note here. First, we only observe a significant number of word detectors at the VQ3 layer when that layer is trained from a cold-start. Even when adding a second VQ layer as in the "$\{3\} \to \{2, 3\}$" model, these detectors remain at VQ3.

Jointly training VQ2 and VQ3 together from a cold-start results in the word detectors being divided between those layers. While this experiment demonstrates that it is possible to jointly train two quantizers at once, the $\{2, 3\}$ model learned the smallest total number of word detectors of any model. Additionally, we were unable to successfully train a cold-start $\{2, 3, 4\}$ model; these experiments suggest that training quantizers one by one may be easier in general.

For all models beginning from a "$\varnothing$" or "$\{2\}$" initial model, we do not observe any word detectors at either VQ2 or VQ3. However, a third quantizer at the VQ4 position in the "$\{2\} \to \{2, 3\} \to \{2, 3, 4\}$" model was able to capture words. We hypothesize that in all models not trained from a "$\{3\}$" initialization, word recognition is implicitly learned by the Res4 layer, and adding a quantizer to the output of this layer serves to make the categorical nature of those representations explicit.

### A.3 WORD DETECTOR TABLES FOR VARIOUS MODELS

In Table 7, we show a sampling of 50 word-detecting codebook entries from the VQ layer of the "$\{3\} \to \{2, 3\}$" model (many word detectors learned). Analogous results for the "$\{2\} \to \{2, 3\}$" model (few word detectors learned) are shown in Table 8.

### A.4 UNIT VISUALIZATION FOR INDIVIDUAL CAPTION SPECTROGRAMS

To provide a better intuitive understanding of what the units learned by our models look like, in Figures 5, 6, and 7, we display speech spectrograms for several Places caption fragments. Along with each spectrogram we display the time-aligned, ground-truth, word-level text (top transcription), the inferred unit sequence for the VQ4 layer (middle transcription), and the unit sequence for the VQ3 (bottom transcription) layer. All VQ unit alignments in these figures are derived from the "$\{2\} \to \{2, 3\} \to \{2, 3, 4\}$" model.

Table 7: Performance of the VQ3 layer from the "$\{3\} \rightarrow \{2, 3\}$" model when codes are treated as word detectors. Codes are ranked by the highest F1 score among the retrieved words for a given code. Word hypotheses for a given code are ranked by the F1 score.

| rank | code | Top Hypotheses | | | | Second Hypotheses | | | |
| | | word | F1 | P | R | occ | word | F1 | P | R | occ |
|---|---|---|---|---|---|---|---|---|---|---|---|
| 1 | 918 | pantry | 90.67 | 88.29 | 93.18 | 41 | spice | 3.96 | 2.20 | 20.00 | 1 |
| 2 | 596 | kitchen | 90.08 | 91.59 | 88.63 | 304 | countertop | 1.64 | 0.84 | 29.63 | 8 |
| 3 | 88 | classroom | 88.97 | 89.05 | 88.89 | 72 | classrooms | 5.01 | 2.57 | 100.00 | 2 |
| 4 | 58 | baseball | 88.71 | 88.63 | 88.78 | 182 | player | 3.01 | 1.65 | 17.11 | 13 |
| 5 | 706 | background | 87.86 | 91.93 | 84.14 | 838 | ground | 0.58 | 0.39 | 1.18 | 4 |
| 6 | 736 | museum | 87.35 | 93.44 | 82.00 | 41 | museums | 5.47 | 2.81 | 100.00 | 1 |
| 7 | 274 | subway | 87.26 | 88.34 | 86.21 | 75 | assembly | 5.32 | 2.85 | 40.00 | 4 |
| 8 | 116 | construction | 87.07 | 89.78 | 84.52 | 131 | constructed | 2.43 | 1.25 | 38.46 | 5 |
| 9 | 892 | walking | 87.06 | 87.57 | 86.55 | 412 | walk | 7.07 | 3.97 | 31.94 | 23 |
| 10 | 557 | concrete | 86.53 | 90.98 | 82.50 | 99 | concur | 1.21 | 0.61 | 100.00 | 1 |
| 11 | 48 | desert | 86.50 | 90.30 | 83.01 | 171 | dozen | 2.76 | 1.49 | 18.18 | 2 |
| 12 | 534 | background | 86.18 | 81.95 | 90.86 | 905 | back | 8.95 | 5.83 | 19.34 | 64 |
| 13 | 44 | patio | 85.82 | 90.87 | 81.29 | 113 | patios | 1.56 | 0.79 | 100.00 | 1 |
| 14 | 625 | background | 85.17 | 92.92 | 78.61 | 783 | back | 1.63 | 1.01 | 4.23 | 14 |
| 15 | 732 | closet | 84.92 | 94.64 | 77.01 | 67 | closets | 4.96 | 2.68 | 33.33 | 2 |
| 16 | 30 | waterfall | 84.90 | 75.73 | 96.61 | 57 | waterfalls | 14.26 | 7.68 | 100.00 | 7 |
| 17 | 388 | courtyard | 84.89 | 92.16 | 78.69 | 48 | graveyard | 5.50 | 3.24 | 18.18 | 4 |
| 18 | 560 | hospital | 84.70 | 91.48 | 78.85 | 41 | horses | 1.55 | 1.92 | 1.30 | 1 |
| 19 | 18 | driveway | 84.56 | 90.10 | 79.66 | 47 | driveways | 3.52 | 1.82 | 50.00 | 1 |
| 20 | 598 | palm | 84.39 | 82.08 | 86.84 | 99 | plum | 1.57 | 0.79 | 100.00 | 1 |
| 21 | 85 | yellow | 84.30 | 83.93 | 84.66 | 574 | yellowish | 1.98 | 1.00 | 100.00 | 7 |
| 22 | 584 | playground | 84.18 | 77.37 | 92.31 | 36 | play | 6.32 | 4.75 | 9.43 | 5 |
| 23 | 162 | stadium | 83.82 | 84.50 | 83.15 | 74 | boardwalk | 9.12 | 4.91 | 63.16 | 12 |
| 24 | 769 | bamboo | 83.79 | 93.68 | 75.79 | 72 | baboons | 2.03 | 1.03 | 100.00 | 1 |
| 25 | 193 | small | 83.55 | 90.46 | 77.63 | 791 | smaller | 2.15 | 1.10 | 50.00 | 15 |
| 26 | 412 | podium | 83.53 | 76.73 | 91.67 | 22 | auditorium | 8.69 | 5.82 | 17.14 | 6 |
| 27 | 108 | highway | 83.52 | 79.58 | 87.88 | 58 | highlights | 5.44 | 2.87 | 50.00 | 3 |
| 28 | 394 | church | 83.34 | 75.98 | 92.28 | 227 | religious | 6.34 | 3.45 | 39.39 | 13 |
| 29 | 661 | distance | 83.32 | 78.96 | 88.19 | 351 | lounge | 1.63 | 0.86 | 14.71 | 5 |
| 30 | 708 | distance | 82.97 | 96.32 | 72.86 | 290 | farmland | 1.35 | 0.68 | 55.56 | 5 |
| 31 | 14 | gallery | 82.97 | 85.08 | 80.95 | 17 | art | 11.39 | 12.15 | 10.71 | 9 |
| 32 | 996 | large | 82.95 | 87.05 | 79.21 | 1753 | very | 2.83 | 1.72 | 8.01 | 94 |
| 33 | 944 | cathedral | 82.78 | 79.22 | 86.67 | 52 | feed | 2.74 | 1.41 | 50.00 | 1 |
| 34 | 122 | purple | 82.63 | 91.98 | 75.00 | 138 | proportion | 1.66 | 0.84 | 50.00 | 1 |
| 35 | 630 | trees | 82.52 | 80.39 | 84.77 | 1258 | tree | 15.48 | 9.47 | 42.33 | 171 |
| 186 | 375 | boy | 69.90 | 65.45 | 75.00 | 93 | boys | 20.87 | 13.09 | 51.43 | 18 |
| 187 | 634 | ground | 69.78 | 73.92 | 66.08 | 224 | playground | 7.45 | 3.94 | 69.23 | 27 |
| 188 | 69 | courtyard | 69.55 | 57.98 | 86.89 | 53 | plaza | 28.89 | 19.87 | 52.94 | 9 |
| 189 | 281 | wooden | 69.50 | 57.89 | 86.92 | 525 | wood | 23.55 | 14.52 | 62.20 | 153 |
| 190 | 812 | lighthouse | 69.41 | 59.74 | 82.81 | 53 | lighthouses | 11.69 | 6.21 | 100.00 | 5 |
| 191 | 225 | house | 69.13 | 61.59 | 78.78 | 516 | houses | 18.29 | 10.31 | 80.73 | 88 |
| 192 | 705 | dark | 69.11 | 68.57 | 69.66 | 186 | darker | 3.05 | 1.56 | 75.00 | 6 |
| 193 | 980 | building | 69.10 | 77.48 | 62.35 | 1161 | buildings | 25.72 | 15.71 | 70.78 | 281 |
| 194 | 844 | grass | 69.01 | 61.85 | 78.04 | 455 | grassy | 21.70 | 12.42 | 85.83 | 109 |
| 195 | 446 | lake | 68.69 | 78.63 | 60.98 | 125 | late | 5.02 | 2.90 | 18.52 | 5 |
| 196 | 182 | trash | 68.64 | 66.79 | 70.59 | 48 | boulders | 10.16 | 6.27 | 26.67 | 4 |
| 197 | 437 | photograph | 68.63 | 59.06 | 81.89 | 588 | photographs | 30.56 | 18.46 | 88.73 | 181 |
| 198 | 237 | lobby | 68.43 | 56.77 | 86.11 | 31 | waiting | 9.93 | 7.86 | 13.46 | 14 |
| 199 | 829 | shirt | 68.41 | 71.49 | 65.58 | 322 | shirts | 18.28 | 10.37 | 76.79 | 43 |
| 200 | 59 | grass | 68.31 | 56.53 | 86.28 | 503 | grassy | 15.30 | 8.67 | 65.35 | 83 |

Table 8: Performance of the VQ3 layer from the "$\{2\} \rightarrow \{2, 3\}$" model when codes are treated as word detectors. Codes are ranked by the highest F1 score among the retrieved words for a given code. Word hypotheses for a given code are ranked by the F1 score.

| rank | code | Top Hypotheses | | | | | Second Hypotheses | | | | |
|------|------|------|-----|-----|-----|-----|------|-----|-----|-----|-----|
| | | word | F1 | P | R | occ | word | F1 | P | R | occ |
| 1 | 924 | people | 76.71 | 67.49 | 88.85 | 1665 | computer | 2.17 | 1.12 | 40.40 | 40 |
| 2 | 749 | white | 76.47 | 66.92 | 89.21 | 2265 | one | 4.15 | 2.50 | 12.14 | 134 |
| 3 | 530 | building | 75.47 | 64.84 | 90.28 | 1681 | buildings | 23.93 | 13.81 | 89.67 | 356 |
| 4 | 505 | blue | 59.12 | 46.90 | 79.96 | 1093 | pool | 10.74 | 5.89 | 60.80 | 152 |
| 5 | 581 | snow | 57.61 | 41.77 | 92.83 | 466 | snowy | 16.63 | 9.12 | 94.50 | 103 |
| 6 | 778 | building | 52.10 | 36.71 | 89.69 | 1670 | buildings | 14.30 | 7.78 | 88.16 | 350 |
| 7 | 144 | with | 49.12 | 41.58 | 59.99 | 3386 | wooden | 6.32 | 3.34 | 60.60 | 366 |
| 8 | 299 | small | 47.83 | 32.78 | 88.42 | 901 | snow | 30.55 | 18.35 | 91.24 | 458 |
| 9 | 550 | large | 45.13 | 30.50 | 86.76 | 1920 | car | 8.57 | 4.52 | 82.13 | 216 |
| 10 | 76 | trees | 44.82 | 29.76 | 90.77 | 1347 | tree | 15.21 | 8.31 | 89.36 | 361 |
| 11 | 831 | water | 41.84 | 27.50 | 87.43 | 1210 | wall | 17.57 | 9.87 | 79.97 | 491 |
| 12 | 1015 | large | 39.59 | 26.06 | 82.29 | 1821 | cars | 6.58 | 3.42 | 84.21 | 224 |
| 13 | 80 | red | 39.15 | 26.13 | 78.05 | 992 | bed | 9.14 | 4.91 | 66.67 | 168 |
| 14 | 719 | woman | 38.71 | 24.95 | 86.31 | 687 | women | 7.75 | 4.09 | 72.41 | 126 |
| 15 | 614 | people | 37.71 | 25.10 | 75.83 | 1421 | table | 22.10 | 12.75 | 82.93 | 656 |
| 16 | 816 | water | 37.71 | 24.25 | 84.75 | 1173 | river | 5.72 | 2.97 | 80.22 | 215 |
| 17 | 457 | sky | 35.71 | 23.20 | 77.47 | 540 | skies | 11.81 | 6.33 | 88.12 | 141 |
| 18 | 480 | has | 34.88 | 25.01 | 57.64 | 1204 | house | 9.88 | 5.48 | 50.38 | 330 |
| 19 | 245 | yellow | 34.14 | 21.17 | 88.05 | 597 | flowers | 13.44 | 7.27 | 89.43 | 237 |
| 20 | 968 | picture | 34.11 | 22.32 | 72.33 | 1686 | pictures | 16.79 | 9.38 | 79.77 | 698 |
| 21 | 985 | trees | 33.88 | 22.04 | 73.25 | 1087 | tree | 10.96 | 5.93 | 72.52 | 293 |
| 22 | 536 | man | 33.53 | 20.71 | 88.06 | 1128 | standing | 9.06 | 4.86 | 66.17 | 532 |
| 23 | 0 | black | 33.49 | 20.81 | 85.77 | 1163 | background | 21.15 | 12.28 | 76.20 | 759 |
| 24 | 815 | with | 33.21 | 22.76 | 61.36 | 3463 | white | 8.60 | 4.85 | 38.05 | 966 |
| 25 | 293 | large | 33.13 | 23.80 | 54.50 | 1206 | bridge | 19.09 | 10.78 | 83.18 | 371 |
| 26 | 870 | trees | 32.87 | 20.79 | 78.44 | 1164 | train | 12.97 | 7.00 | 88.03 | 353 |
| 27 | 153 | yellow | 32.42 | 19.94 | 86.73 | 588 | area | 15.39 | 9.47 | 40.92 | 354 |
| 28 | 243 | front | 32.13 | 20.97 | 68.69 | 895 | from | 14.34 | 8.51 | 45.49 | 358 |
| 29 | 538 | black | 31.40 | 19.64 | 78.24 | 1061 | glass | 8.28 | 4.36 | 82.90 | 223 |
| 30 | 526 | small | 31.34 | 19.08 | 87.83 | 895 | large | 5.91 | 4.04 | 11.03 | 244 |
| 31 | 395 | picture | 31.32 | 20.90 | 62.46 | 1456 | pictures | 13.98 | 7.80 | 67.54 | 591 |
| 32 | 133 | white | 29.82 | 18.98 | 69.55 | 1766 | black | 16.34 | 9.32 | 66.37 | 900 |
| 33 | 715 | white | 29.45 | 19.73 | 58.05 | 1474 | like | 9.92 | 5.89 | 31.29 | 388 |
| 34 | 39 | picture | 29.37 | 22.51 | 42.26 | 985 | like | 9.82 | 5.79 | 32.26 | 400 |
| 35 | 740 | trees | 29.31 | 18.52 | 70.35 | 1044 | showcasing | 3.88 | 2.00 | 61.51 | 155 |
| 186 | 129 | area | 6.62 | 3.69 | 32.37 | 280 | snow | 6.08 | 3.23 | 51.20 | 257 |
| 187 | 288 | picture | 6.61 | 3.76 | 27.63 | 644 | pictures | 4.91 | 2.58 | 52.91 | 463 |
| 188 | 374 | there's | 6.61 | 3.73 | 28.71 | 775 | that | 5.14 | 2.91 | 21.81 | 614 |
| 189 | 1003 | front | 6.59 | 3.65 | 34.00 | 443 | some | 4.56 | 2.64 | 16.84 | 345 |
| 190 | 396 | multi | 6.45 | 6.25 | 6.67 | 1 | towers | 5.41 | 6.25 | 4.76 | 1 |
| 191 | 790 | photo | 6.23 | 3.35 | 44.27 | 247 | by | 5.66 | 3.06 | 37.60 | 229 |
| 192 | 522 | field | 6.14 | 3.22 | 63.22 | 385 | photo | 6.01 | 3.19 | 51.79 | 289 |
| 193 | 42 | man | 5.96 | 3.19 | 45.12 | 578 | middle | 3.85 | 1.98 | 68.11 | 314 |
| 194 | 801 | sitting | 5.91 | 3.15 | 48.56 | 405 | with | 5.00 | 3.65 | 7.92 | 447 |
| 195 | 181 | trees | 5.89 | 3.16 | 43.19 | 641 | with | 5.13 | 3.22 | 12.72 | 718 |
| 196 | 791 | with | 5.84 | 3.58 | 15.86 | 895 | that | 3.21 | 1.79 | 15.88 | 447 |
| 197 | 975 | purple | 5.72 | 2.99 | 64.13 | 118 | parked | 4.67 | 2.42 | 65.93 | 120 |
| 198 | 38 | structure | 5.62 | 6.20 | 5.14 | 13 | graph | 4.03 | 2.19 | 25.00 | 2 |
| 199 | 432 | parking | 5.61 | 2.90 | 83.12 | 133 | park | 5.56 | 2.87 | 82.89 | 126 |
| 200 | 329 | standing | 5.60 | 2.97 | 49.63 | 399 | woman | 3.09 | 1.62 | 34.67 | 276 |

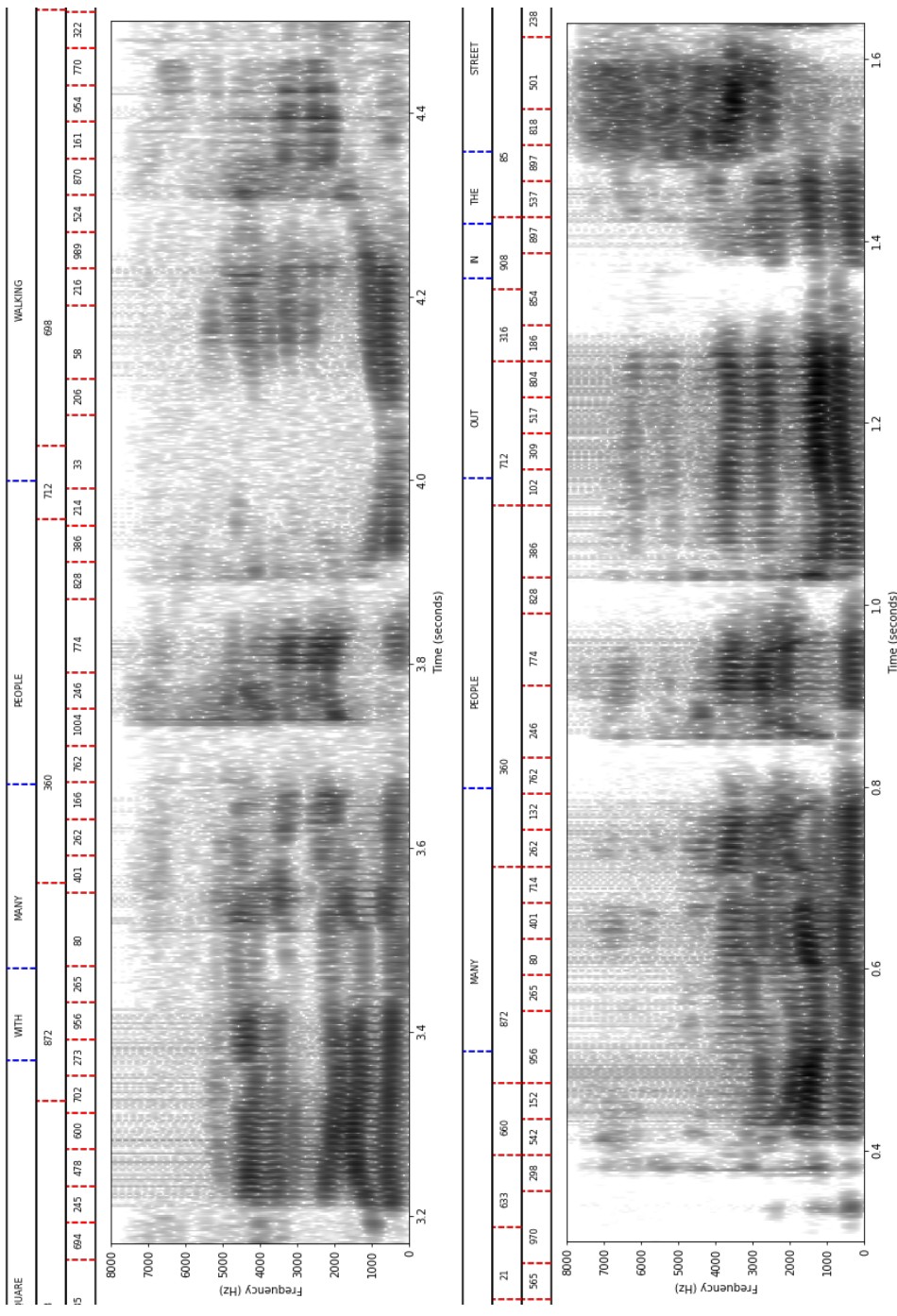

Figure 5: Two different captions containing the phrase "many people." In both cases, the VQ4 layer infers the same unit sequence (872, 360, 712, middle transcription) beneath the phrase. The VQ3 units are somewhat noisier, but contain the common subsequence (956, 265, 80, 401, 262, 762, 246, 774, 828, 386, bottom transcription).

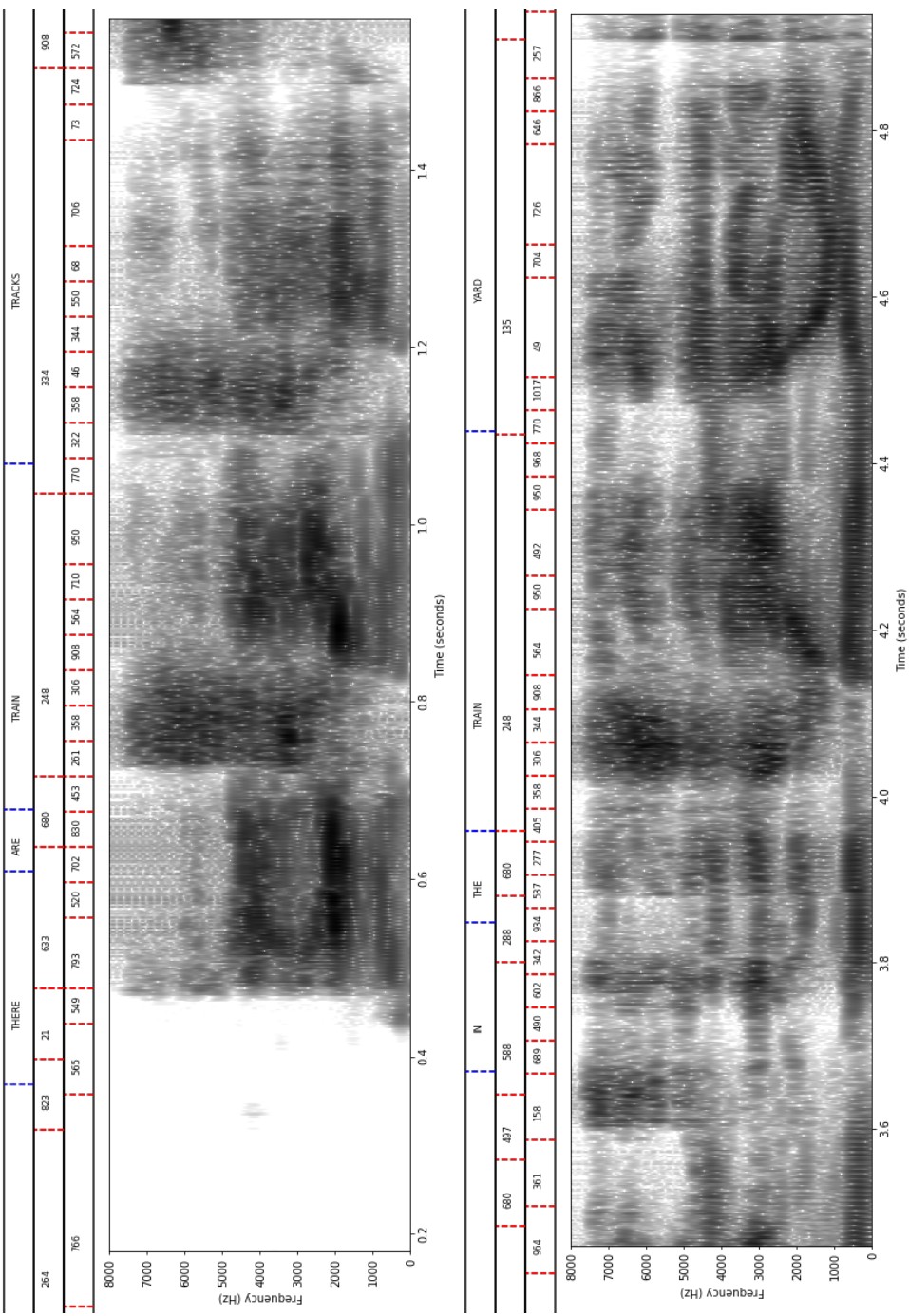

Figure 6: Two different captions containing word "train". In both cases, the VQ4 layer infers the same unit sequence (680, 248, top transcription) surrounding the word "train". The VQ3 alignments contain the same common subsequence (358, 306, 908, 564, 950, 770, bottom transcription). Notice that the same (358, 306) VQ3 unit sequence is aligned to the /tr/ phone cluster at the beginning of both instances of the word "train," as well as the /tr/ at the beginning of both instances of "tree" in Figure 7. Unit 358 is also found covering the /tr/ at the beginning of the word "tracks" in the topmost spectrogram (although unit 306 is absent in this case).

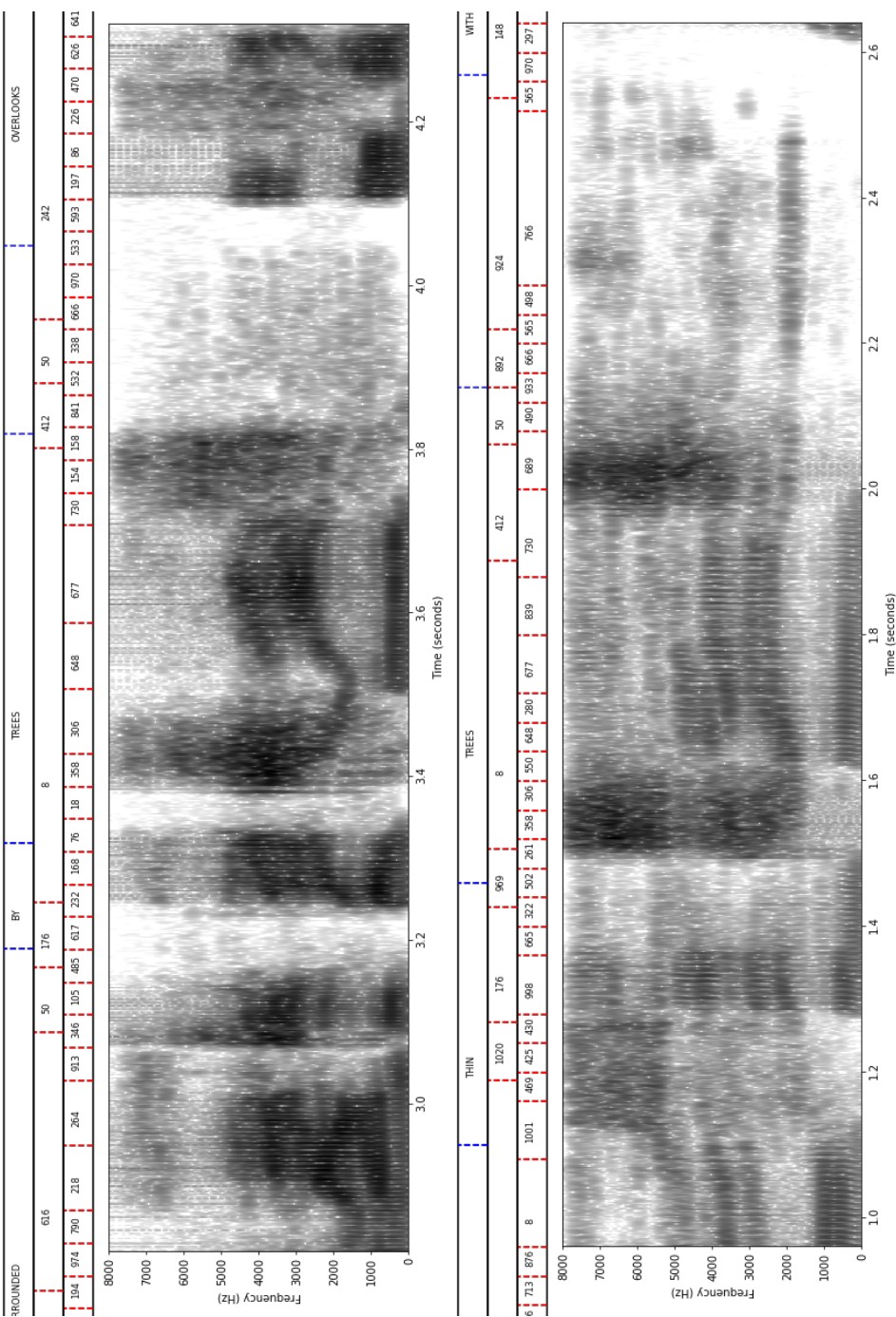

Figure 7: Two different captions containing word "trees". In both cases, the VQ4 layer infers the same unit sequence (8, 412, 50, top transcription) surrounding the word "trees". The VQ3 alignments contain the same common subsequence (358, 306, 648, 677, 730, bottom transcription). Notice that the (358, 306) VQ3 unit sequence is aligned to the /tr/ phone cluster at the beginning of both instances of "trees," and this same unit sequence is inferred for the /tr/ phone sequence in both instances of "train" in Figure 6.

