# OpenReview forum: "Learning Hierarchical Discrete Linguistic Units from Visually-Grounded Speech"
_ICLR.cc/2020/Conference — Accept (Talk)_

### Official Review · AnonReviewer1 · 2019-10-21
**Official Blind Review #1**

**Rating:** 8

**Review:**

Overview:

The paper proposes a method to learn discrete linguistic units in a low-resource setting using speech paired with images (no labels). The visual grounding signal is different from other recent work, where a reconstruction objective was used to learn discrete representations in unsupervised neural networks. In contrast to other work, a hierarchy of discretization layers are also considered, and the paper shows that, with appropriate initialization, higher discrete layers capture word-like units while lower layers capture phoneme-like units.

Strengths:

The paper is extremely well-written with a clear motivation (Section 1). The approach is novel. But I think the paper's biggest strength is in its very thorough experimental investigation. Their approach is compared to other very recent speech discretization methods on the same data using the same (ABX) evaluation metric. But the work goes further in that it systematically attempts to actually understand what types of structures are captured in the intermediate discrete layers, and it is able to answer this question convincingly. Finally, very good results on standard benchmarks are achieved.

Weaknesses:

Although I think the paper is very well-motivated, my first criticism is that discretization itself is not motivated: why is it necessary to have a model with discrete intermediate layers? Does this give us something other than interpretability (which we obtain due to the sparse bottleneck)? In the detailed questions below, I also specifically ask whether, for instance, the downstream speech-image task actually benefits from including discrete layers.

My second point is that it is unclear why word-like units only appear when the higher-level discrete layers are trained from scratch; as soon as warm-starting is used, the higher level layers capture phoneme-like units (Table 1). Is it possible to answer/speculate why this is the case?

Overall assessment:

The paper presents a new approach with a thorough experimental investigation. I therefore assign an "accept". The weaknesses above asks for additional motivation and some speculation.

Questions, suggestions, typos, grammar and style:

- Section 3.3: It maybe makes less sense for the end-task, but did the authors consider discretization on the image side of the network? This could maybe lead to parts of objects being composed to form larger objects (in analogy to the speech network).
- Section 3.3, par. 3: "with the intention that they should capture discrete word-like and sub-word-like units" -> "with the intention that they should capture discrete *sub-word-like and word-like units*" (easier to read with first part of sentence)
- Section 3.3: The more standard VQ-VAE adds a commitment loss and a loss for updating the embeddings; was this used or considered at all, or is this all captured through the exponential moving average method?
- Section 3.4: "with same VQ layers" -> "with *the* same VQ layers"
- Section 3.5: Can you briefly outline the motivation for adding the two losses (so that it is not required to read the previous work).
- Section 4.1: Following from the first weakness listed above, the caption under Figure 2 states that the non-discrete model achieves a speech-image retrieval R@10 of 0.735. This is lower than some of the best scores achieved in Table 1. Can this be taken as evidence that discretization actually improves the downstream task? If so, it would be worth highlighting the point more; if there is some other reason, that would also be worth knowing.
- Figure 1: Did the authors ever consider putting discrete layers right at the top of the speech component, just before the pooling layer? Would this more consistently lead to word-like units?


**Experience Assessment:**

I have published in this field for several years.

**Review Assessment: Checking Correctness Of Derivations And Theory:**

I assessed the sensibility of the derivations and theory.

**Review Assessment: Checking Correctness Of Experiments:**

I carefully checked the experiments.

**Review Assessment: Thoroughness In Paper Reading:**

I read the paper thoroughly.

---

> ### Author Response · Authors · 2019-11-08
> **Response to comments from Reviewer #1 (part 2 of 2)**
>
> Detailed response to Reviewer #1 (part 2 of 2):
>
> Q1.6: My second point is that it is unclear why word-like units only appear when the higher-level discrete layers are trained from scratch; as soon as warm-starting is used, the higher level layers capture phoneme-like units (Table 1). Is it possible to answer/speculate why this is the case?
>
> A1.6: We were equally intrigued as to the reason why the training curriculum had such a large impact on which level of the linguistic hierarchy (phones vs. words) was captured by a particular discretization layer. We have continued to perform experiments after the initial submission of the paper, and have found that when we take a quantized model that did not learn word detectors and add another quantization layer to it after the 4th layer (in the notation used in our paper this would correspond to {2} -> {2,3} -> {2,3,4}), the VQ4 layer did in fact learn to capture words. Thus, it seems that warm-start training pushes the emergence of words higher up into the network, whereas cold-start training with the 3rd layer quantized can pull the word detectors down into the middle of the network.
>
> We hypothesize that the phenomenon results from two factors: (1) for warm-start models, the type of information encoded at each layer does not change after adding a quantization layer and fine-tuning the model, and (2) for cold-start models, adding a quantization layer effectively forms a bottleneck at that layer, restricting the amount of information allowed to pass through that layer. Thus, if word-level recognition tends to emerge at the 4th residual layer in an unconstrained network with no quantization bottleneck, it will stay there when quantization is introduced for fine-tuning. However, when a quantization bottleneck is introduced at the 3rd residual layer from the very beginning of training, the gradient flowing down into the lower network layers is far more constrained during the initial training epochs (when the gradient tends to be the largest). This may have the effect of steering the optimizer into a very different part of the parameter space, where word-level recognition is localized at the 3rd residual layer.
>
> The first point can be empirically verified through evaluating the ABX score at layer 3 for models with the same initial model. The results are shown below.
>
> Model          | ABX @ layer-3 (pre-quantized)
> ————————————————————————————
> {}                  |   10.86%
> {}->{2}         |   11.61%
> {}->{3}         |   10.91%
> {}->{2,3}      |   12.68%
> ————————————————————————————
> {2}                |   11.45%
> {2}->{2,3}    |   11.37%
> ————————————————————————————
> {3}                |   32.24%
> {3}->{2,3}    |   28.33%
>
> In light of our recent finding that word-level recognition emerges at the 4th layer in warm-start models, we plan to update the paper with those new results, which we believe will tell a more complete story about how the training curriculum influences our hierarchical discretization method.
>
> Q1.7: - Section 4.1: Following from the first weakness listed above, the caption under Figure 2 states that the non-discrete model achieves a speech-image retrieval R@10 of 0.735. This is lower than some of the best scores achieved in Table 1. Can this be taken as evidence that discretization actually improves the downstream task? If so, it would be worth highlighting the point more; if there is some other reason, that would also be worth knowing.
>
> A1.7: This is exactly right. Except for “{3}” and “{2,3}” (quantizing layer 2 and quantizing layer 2&3 from scratch), all the discrete models improve the visual grounding task compared to the non-discrete model. This can be interpreted as incorporating the prior knowledge that linguistic units are inherently discrete at many different levels (e.g., phonemes and words) into the model architecture improves generalization on the downstream task. However, we also want to note that the best representations extracted from the non-discrete model still achieves a better performance on ABX evaluation compared to those from the discrete model (10.86 vs 11.79). We will incorporate the reviewer’s comment and highlight the visual grounding improvement in our paper.
>
> Writing Suggestions:
> - Section 3.3, par. 3: "with the intention that they should capture discrete word-like and sub-word-like units" -> "with the intention that they should capture discrete *sub-word-like and word-like units*" (easier to read with first part of sentence)
> - Section 3.4: "with same VQ layers" -> "with *the* same VQ layers"
>
> A: We appreciate the additional suggestions and will incorporate them into our edits.

---

> > ### Comment · AnonReviewer1 · 2019-11-14
> > **Thanks!**
> >
> > Thank you for very thorough answers to all my questions, and for the substantial additional experiments which further strengthens the paper.

---

> ### Author Response · Authors · 2019-11-08
> **Response to comments from Reviewer #1 (part 1 of 2)**
>
> Detailed response to Reviewer #1 (part 1 of 2):
>
> Q1.1: why is it necessary to have a model with discrete intermediate layers? Does this give us something other than interpretability (which we obtain due to the sparse bottleneck)? In the detailed questions below, I also specifically ask whether, for instance, the downstream speech-image task actually benefits from including discrete layers.
>
> A1.1:
> We agree that interpretability is an important benefit of quantization, and also believe that there are other compelling reasons for learning discrete speech representations.
>
> All known human languages employ a finite inventory of phonemes, where a phoneme is defined as the minimal perceptive unit whose modification (insertion, deletion, substitution) changes the meaning of the underlying word it belongs to. The sentences “They bit them” and “They bet them” differ by only one phoneme (/I/ vs. /e/ in the middle word), but as a result they take on completely different meanings. Evidence has been found that human perception of phonemes is categorical in nature: for example, gradually interpolating between the acoustic realizations of two different phonemes reveals a sharp decision threshold in human judgements of the underlying phoneme identity [1].
>
> From the perspective of one-shot or few-shot learning, discretization makes sense because the principle of compositionality could be leveraged to reduce the sample complexity of learning new words. From a bottom-up standpoint, maintaining an inventory of sub-word building blocks enables new words to be expressed in terms of a sequence of units already known to the model. From a top-down perspective, segmentation and grammatical parsing of the words surrounding a new word form enable a learner to more quickly grasp the meaning of the word. Ultimately, we would like to extend this work into the realm of acquiring higher-level linguistic structure, such as syntax and grammar, directly from the speech waveform (as human children are able to do). These systems inherently operate on token sequences, and so it seems reasonable that some sort of discrete inductive bias should be baked into the model.
>
> Finally, learning discrete representations of speech audio opens the door to applying NLP techniques such as BERT directly to the speech signal. This was explored in another ICLR 2020 submission, with impressive results on supervised speech recognition (https://openreview.net/forum?id=rylwJxrYDS)
>
> [1] Lisker, Leigh, and Arthur S. Abramson. "The voicing dimension: Some experiments in comparative phonetics." In Proceedings of the 6th international congress of phonetic sciences, pp. 563-567. Academia Prague, 1970.
>
> Q1.2: Section 3.3: It maybe makes less sense for the end-task, but did the authors consider discretization on the image side of the network? This could maybe lead to parts of objects being composed to form larger objects (in analogy to the speech network).
>
> A1.2: We have not experimented with adding VQ layers in the image model, but we agree with the reviewer that it may also lead to a similar hierarchical compositionality in the image network, and further improves the visual grounding performance because of better generalization, as observed when adding VQ layers to the audio network (as noted in Q1.6). We will leave it as future work here.
>
> Q1.3: - Section 3.3: The more standard VQ-VAE adds a commitment loss and a loss for updating the embeddings; was this used or considered at all, or is this all captured through the exponential moving average method?
>
> A1.3: We only experimented with exponential moving average (EMA) for updating the codebook, which is introduced in the original VQ-VAE paper and works well for our model. We believe that our model should also work with the gradient-based update.
>
> Q1.4: - Section 3.5: Can you briefly outline the motivation for adding the two losses (so that it is not required to read the previous work).
>
> A1.4: Yes, we will add the motivation for adding the losses to the text. It was shown in Harwath et al. (2019) that the addition of the second loss term (semi-hard negative mining) performed much better than the standard triplet loss, and the authors noted that using only the semi-hard negative mining term by itself led to unstable training.
>
> Q1.5:- Figure 1: Did the authors ever consider putting discrete layers right at the top of the speech component, just before the pooling layer? Would this more consistently lead to word-like units?
>
> A1.5: We did not consider adding a discrete layer because we hypothesize that the final layer could be capturing linguistic units larger than words, such as phrases (which would require a much larger codebook), or semantic equivalence classes of multiple different words/phrases with synonymous/similar meanings.

---

### Official Review · AnonReviewer2 · 2019-10-22
**Official Blind Review #2**

**Rating:** 8

**Review:**

This paper attempts to learn discrete speech units in a hierarchical (phone and word) fashion by incorporating multiple vector quantization layers into the audio encoder branch of a model that visually grounds speech segments with accompanying images.

The model has been tested and compared against two algorithms and implementations that set the SOTA on the Zero Speech 2019 challenge (further improving one of them in the process, it seems), and outperforms these significantly using the ABX metric, so the proposed method seems to perform well (the model is using additional supervision, though). In addition, this is an interesting and timely research problem with implications far beyond the core machine learning setup. The hierarchical setup, and the finding that successful learning here depends on the curriculum, is intriguing indeed. The paper is a pleasure to read and provides a rich set of results and analyses.

A few remarks:
- It is probably worth explaining how the ABX test is performed, i.e. that features are extracted from some layer of a model, and then a time alignment is performed to get the score - this is in the text somehow, but i had to read it multiple times.
- Did you try other architectures like 5 layers (rather than 4) in Figure 2
- Figure 2 is a bit hard to interpret. Maybe plot log of ABX error rate or something, to pull apart the different layers?
- Could you explain the difference between cold-start and warm-start? One is adding the discretization to a pre-trained model, the other is training from the start?
- When you measure ABX at layer 2 and 3, in a model trained with quantization, do you measure ABX on the features before or after quantization? does it make a difference?
- Table 7: some of the top word hypothesis pairs make sense acoustically (building-buildings, red-bed, ...), some could be neighboring words (large-car, ...), but some are just weird (people-computer) - any intuition as to what is going on?

**Experience Assessment:**

I have published one or two papers in this area.

**Review Assessment: Checking Correctness Of Derivations And Theory:**

I assessed the sensibility of the derivations and theory.

**Review Assessment: Checking Correctness Of Experiments:**

I assessed the sensibility of the experiments.

**Review Assessment: Thoroughness In Paper Reading:**

I read the paper at least twice and used my best judgement in assessing the paper.

---

> ### Author Response · Authors · 2019-11-08
> **Response to comments from Reviewer #2**
>
> Detailed response to Reviewer #2:
>
> Q2.1: It is probably worth explaining how the ABX test is performed, i.e. that features are extracted from some layer of a model, and then a time alignment is performed to get the score - this is in the text somehow, but i had to read it multiple times.
>
> A2.1: Thank you for this suggestion - we will include a few sentences describing in detail how the ABX score is computed.
>
> Q2.2: Did you try other architectures like 5 layers (rather than 4) in Figure 2
>
> A2.2: We apologize that the legend in Figure 2 is slightly misleading, because “layer” in this context actually refers to the output of a residual block. The total number of convolutional layers in the model is 17 (because each residual block contains 4 convolutions, and there is 1 initial convolutional layer before the 4 residual blocks), but we only probe the representations between consecutive blocks.
>
> Q2.3: Figure 2 is a bit hard to interpret. Maybe plot log of ABX error rate or something, to pull apart the different layers?
>
> A2.3: We agree that the figure would be more clearly interpretable on a log scale and will make this adjustment. We will also make the legend more specific to highlight the fact that in the middle sub-figure the red curve represents a quantized output, and in the right sub-figure the brown curve represents a quantized output (whereas all outputs are continuous in the leftmost sub-figure)
>
> Q2.4: Could you explain the difference between cold-start and warm-start? One is adding the discretization to a pre-trained model, the other is training from the start?
>
> A2.4: A warm-start model just means that training starts from an initial model that can either be continuous or discretized - in other words, a warm-start model is pre-trained, except for any new quantization layers that are inserted (which are randomly initialized). In contrast, a cold-start model is trained from a random initialization.
>
> Q2.5: When you measure ABX at layer 2 and 3, in a model trained with quantization, do you measure ABX on the features before or after quantization? does it make a difference?
>
> A2.5: In these cases, we measure ABX on the features after quantization in the paper. Here we provide additional ABX results on the features before quantization.
>
> For model “{2}” in Table 1, ABX at layer 2 is 10.73% before quantization and 12.33% after quantization. For model “{3}” in Table 1, ABX at layer 3 is 32.24% before quantization and 38.21% after quantization. In general, quantization does hurt the ABX performance, but comes with the benefit of far greater information compression (in terms of bitrate) over non-quantized representations.  Additionally, it should also be noticed that for non-quantized model, the ABX at layer 2 is 11.35%, which is worse than the pre-quantized feature in a quantized model. This can imply that the inductive bias that encourages features to be close to one of the codes also improves the quality of the learned continuous representations.
>
> Q2.6: Table 7: some of the top word hypothesis pairs make sense acoustically (building-buildings, red-bed, ...), some could be neighboring words (large-car, ...), but some are just weird (people-computer) - any intuition as to what is going on?
>
> A2.6: Table 7 represents the performance of the VQ3 layer in a model that did not learn to make this layer function as a word detector. Therefore, we expect very few of the codebook vectors shown in Table 7 to accurately detect words. We included this table primarily as a point of contrast to Table 6, which shows a model whose VQ3 layer did learn to function as a word detector (and thus has many more codes with a large F1 score for specific words). We wanted to draw this contrast because the only difference between the models corresponding to Table 6 vs. Table 7 is the training curriculum. In the case of (people-computer) in Table 7, code #924 was one of only a handful of codebook entries that learned to behave as a reliable word detector, for the word “people” with an F1 score of 76.6. “Computer” is simply the word with the second highest F1 score when code #924 is treated as a detector for that word - but it is a very poor detector for “computer,” with an F1 score of only 2.3.

---

### Official Review · AnonReviewer3 · 2019-10-30
**Official Blind Review #3**

**Rating:** 6

**Review:**

Pretty interesting paper attempting to learn discrete linguistic units via vector quantization of visually grounded, speech related features. I think this is a worthwhile contribution. My main complaint is that the exposition is a bit diffuse and fails to crystallize the essence of the work. In particular, the claim is that the novelty is from the use of a "discriminative, multi-modal grounding objective". Reading the work, this seems to be the triplet loss described in Section 3.5. Is that the novel objective? In my my the really interesting aspect that should be stressed is the visual grounding -- I encourage the authors to highlight that aspect more directly. I fear that essential and interesting point is somewhat diluted in the detailed exposition of results.

**Experience Assessment:**

I have read many papers in this area.

**Review Assessment: Checking Correctness Of Derivations And Theory:**

I assessed the sensibility of the derivations and theory.

**Review Assessment: Checking Correctness Of Experiments:**

I assessed the sensibility of the experiments.

**Review Assessment: Thoroughness In Paper Reading:**

I read the paper at least twice and used my best judgement in assessing the paper.

---

> ### Author Response · Authors · 2019-11-08
> **Response to comments from Reviewer #3**
>
> Detailed response to Reviewer #3:
>
> Q3.1: My main complaint is that the exposition is a bit diffuse and fails to crystallize the essence of the work. In particular, the claim is that the novelty is from the use of a "discriminative, multi-modal grounding objective". Reading the work, this seems to be the triplet loss described in Section 3.5. Is that the novel objective?
>
> A3.1: We apologize for the confusion about the novelty. We did not mean to say the triplet loss is a novel objective for visual grounding. Instead, our novelty is learning discrete linguistic units through visual grounding, rather than through reconstruction (e.g., VQ-VAE). The triplet loss is simply the specific way that we implement the visual grounding.
>
> Q3.2: In my my the really interesting aspect that should be stressed is the visual grounding -- I encourage the authors to highlight that aspect more directly. I fear that essential and interesting point is somewhat diluted in the detailed exposition of results.
>
> A3.2: We thank you for the suggestion and we fully agree. We will revise the abstract and introduction to better highlight discrete speech unit learning via visual grounding as the core contribution.

---

### Author Response · Authors · 2019-11-08
**A thank you and reponse to reviewers**

We would like to thank all three reviewers for taking the time to offer thoughtful and constructive insights, comments, and suggestions. We have replied to each reviewer’s comments individually below.

---

### Decision · Program_Chairs · 2019-12-19

**Decision:**

Accept (Talk)

**Comment:**

The paper is extremely well-written with a clear motivation (Section 1). The approach is novel. But I think the paper's biggest strength is in its very thorough experimental investigation. Their approach is compared to other very recent speech discretization methods on the same data using the same (ABX) evaluation metric. But the work goes further in that it systematically attempts to actually understand what types of structures are captured in the intermediate discrete layers, and it is able to answer this question convincingly. Finally, very good results on standard benchmarks are achieved.

To authors: Please do include the additional discussions and results in the final paper.